# ATTENTION-LIKELIHOOD RELATIONSHIP IN TRANSFORMERS

**Valeria Ruscio** *       **Valentino Maiorca**       **Fabrizio Silvestri**

Sapienza University of Rome

## ABSTRACT

We analyze how large language models (LLMs) represent out-of-context words, investigating their reliance on the given context to capture their semantics. Our likelihood-guided text perturbations reveal a correlation between token likelihood and attention values in transformer-based language models. Extensive experiments reveal that unexpected tokens cause the model to attend less to the information coming from themselves to compute their representations, particularly at higher layers. These findings have valuable implications for assessing the robustness of LLMs in real-world scenarios. Fully reproducible codebase at https://github.com/Flegyas/AttentionLikelihood.

## 1 INTRODUCTION

Transformers, introduced by Vaswani et al. (2017), are the state-of-the-art architecture for language models that rely on self-attention to capture the contextual relationships between tokens in a sentence. While previous research has explored various aspects of the attention mechanism in transformers, such as specialized attention heads and the informative nature of attention weights (Vig & Belinkov, 2019; Kovaleva et al., 2019; Clark et al., 2019), other studies inquire whether attention alone can fully explain the transformer's output (Wiegreffe & Pinter, 2019; Jain & Wallace, 2019; Bibal et al.). Despite this previous work, there is still much to learn about the practical workings of the attention mechanism in transformers. In this study, we investigate the impact of token likelihood, i.e., the modeled probability distribution of a token, on the self-attention mechanism in transformers. Our findings show that tokens with a higher likelihood value receive a correspondingly higher attention value, indicating that the model is relying on the token itself to understand its semantics. However, when out-of-context (i.e., low-likelihood) tokens are encountered, the model redirects its attention to the surrounding context *uniformly*, suggesting that the attention mechanism in transformers handles outliers by nullifying their information. These insights can foster the development of more robust and accurate language models, enhancing their ability to handle unexpected input.

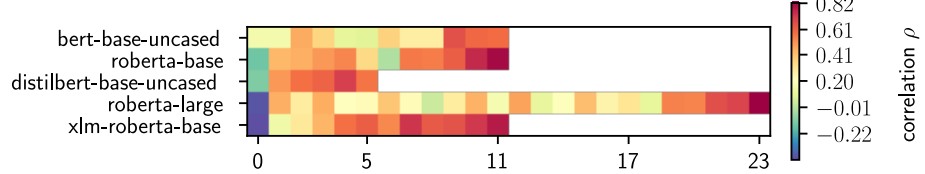

Figure 1: The correlation $\rho$ between likelihood and token attention for various models and layers.

## 2 METHODOLOGY

We conduct a series of likelihood-guided perturbation experiments to investigate the impact of a token's likelihood on the self-attention mechanism in transformers. Let $M$ be a frozen autoencoder transformer model (e.g., BERT (Devlin et al., 2019)), and let $\mathcal{S}$ be a set of sentences. We are

---

*Corresponding author: valeria.ruscio@uniroma1.it

Table 1: The table shows, for both datasets, the mean: token attention $\mathbf{A_{ii}}$, correlation $\rho$ and likelihood $L_s$. The Mann-Whitney U (MWU) is tested between the sentence attentions of the two.

| encoder | layer | Original ($\mathcal{S}$) | | | Perturbed ($\hat{\mathcal{S}}$) | | | MWU |
|---|---|---|---|---|---|---|---|---|
| | | $\bar{\mathbf{A}}_{\mathbf{ii}}$ | $\bar{L}_s$ | $\bar{\rho}$ | $\bar{\hat{A}}_{ii}$ | $\bar{\hat{L}}_s$ | $\bar{\hat{\rho}}$ | p-value |
| BERT | 9 | 0.075 | 0.664 | 0.095 | 0.063 | 0.096 | 0.643 | 0.52 |
| RoBERTa base | 11 | 0.124 | 0.938 | 0.247 | 0.043 | 0.072 | 0.802 | 0.54 |
| XLM-R | 11 | 0.132 | 0.943 | 0.332 | 0.051 | 0.017 | 0.767 | 0.48 |
| RoBERTa large | 23 | 0.134 | 0.947 | 0.183 | 0.046 | 0.055 | 0.823 | 0.56 |
| DistilBERT | 4 | 0.065 | 0.785 | 0.033 | 0.045 | 0.017 | 0.688 | 0.55 |

interested in two different outputs of $M$ applied to each $s \in \mathcal{S}$: **i)** the likelihood $L_{s_i}$ of token $s_i$ at position $i$ in $s$ as the conditional probability $P(s_i|s, M)$; **ii)** the attention matrices $\mathbf{A}^{(l)}$ for each multi-head attention layer $l$ of $M$. Note that we aggregate the outputs of the multiple attention heads in each layer into a single one via mean pooling. After computing likelihood and attention on the original set $\mathcal{S}$, we formulate a likelihood-based perturbation on its sentences as follows: for each sentence $s \in S$, we select a token $s_i$ and replace it with a "perturbed" token $\hat{s_i}$, such that $L_{\hat{s_i}} \ll L_{s_i}$. We then apply $M$ on this perturbed set $\hat{\mathcal{S}}$ to generate the perturbed likelihoods and attention matrices $\hat{\mathbf{A}}^{(l)}$. We define the *token attention* of the token $s_i$ at layer $l$ as $\mathbf{A}_{ii}^{(l)}$. Similarly, the *sentence attention* is the concatenation of all the other attention weights on the $i$-th row of $\mathbf{A}$: $\oplus_{i \neq j} \mathbf{A}_{i,j}^{(l)}$. We compute them for both the original sentences $\mathcal{S}$ and the perturbed ones $\hat{\mathcal{S}}$. This methodology enables us to investigate the impact of a token's likelihood on the self-attention mechanism in transformers and probe the model's ability to adapt to unexpected input at its different layers while maintaining a controlled experimental setup by perturbing each sentence once and only one token per sentence.

## 3 Experiments

We conduct experiments on a dataset $\mathcal{S}$ consisting of around 20k English sentences extracted from WordNet (Miller, 1992). Our analysis involves five transformer models, namely RoBERTa (Liu et al., 2019b) (*base* and *large* versions), BERT (Devlin et al., 2019) and DistilBERT (Sanh et al., 2019) (*base uncased* versions), and XLM-R (Conneau et al., 2020). To investigate the relationship between token likelihood and token attention, we use the Spearman correlation metric on both the original and the perturbed dataset. We report the results of our experiments in table 1. The complete version can be found in the appendix along with additional dataset details. Our findings reveal a **strong correlation** between token likelihood and attention values. Specifically, the token's attention decreases significantly when its likelihood is lowered, indicating that when the model encounters an unexpected token, it shifts its focus from the token to the context to compute its representation. We observe that the attention previously directed to the token is then distributed uniformly to the rest of the sentence, as confirmed by the results of the Mann-Whitney U test (McKnight & Najab, 2010) applied between the attention distributions. Furthermore, we observe that the attention changes due to likelihood-guided perturbations are relevant across most model layers. However, the correlation values are stronger in the latest layers that encode more abstract semantic features Rogers et al. (2020); Liu et al. (2019a). In contrast, in the first layers, the model relies more on the token itself to understand its meaning (Vig & Belinkov, 2019), even when perturbed.

## 4 Conclusion

Our study thoroughly analyzed the link between token likelihood and the self-attention mechanism in transformers, finding a strong statistical correlation between them. This can be seen as a transformer strategy to handle out-of-context words since it redirects its focus from the unlikely token to the context to grasp its semantics, especially at deeper attention layers. Future research could explore the implications of these findings on downstream tasks, enhance language models' robustness, and investigate this phenomenon in different domains, such as ViTs (Dosovitskiy et al., 2020). This could further our understanding of the self-attention mechanism and its application to other fields.

ACKNOWLEDGEMENTS

This work is supported by the ERC Grant no.802554 "SPECGEO" and PRIN 2020 project no.2020TA3K9N "LEGO.AI"

URM STATEMENT

The first author (VR) meets the URM criteria of ICLR 2023 Tiny Papers Track.

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

## A  APPENDIX

### A.1  IMPLEMENTATION DETAILS

We follow the standard best practices for our codebase, both for deep learning and for Python development in general. These are the core tools we relied on:

- *Transformers by HuggingFace* (Wolf et al., 2020), to get ready-to-use transformer-based LLMs;
- *Datasets by HuggingFace* (Lhoest et al., 2021), to handle the creation and processing of our dataset;
- *DVC* (Kuprieiev et al., 2023), for versioning our datasets and results accordingly;
- *NLTK* (Bird et al., 2009) for easy access to WordNet.

Table 2: The HuggingFace transformers used in Section 3, along with some of their specifics.

| official name | Num. Layers | Num. Params |
|---|---|---|
| roberta-base | 12 | 125M |
| bert-base-uncased | 12 | 110M |
| roberta-large | 24 | 355M |
| xlm-roberta-base | 12 | 125M |
| distilbert-base-uncased | 6 | 66M |

### A.2  DATA

To produce the dataset we utilized WordNet. For each synset in it, we retrieved all the lemmas and all the examples. Then, we kept only the examples having an exact match with at least one of the lemmas for their synset. We tokenized the remaining sentences and discarded the ones shorter than 5 tokens. With this procedure, we ended up with a total of 20285 sentences with the length distribution shown in Figure 2.

This was done to ensure both high-quality sentences (WordNet is manually annotated) and that the target token for the perturbation is a "meaningful" one and not a stopword (which could've added negative biases).

### A.3  FULL RESULTS

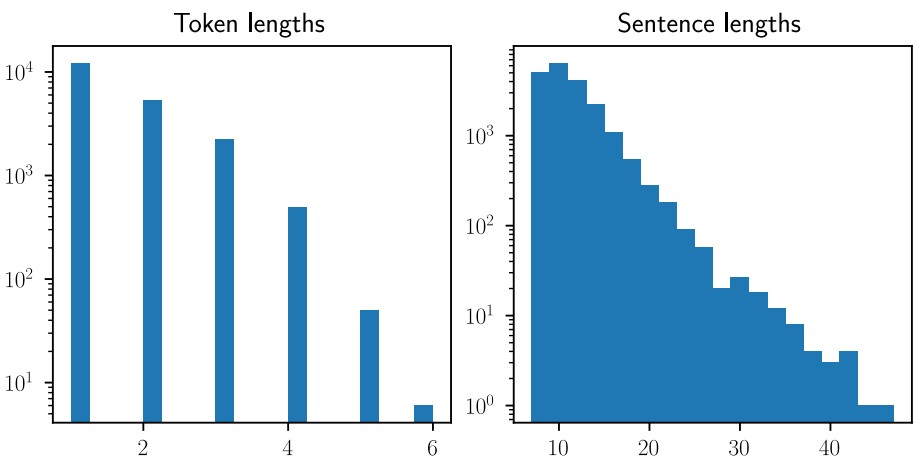

Figure 2: The distributions of the perturbed token lengths (*left*) and the sentence lengths (*right*). Both are calculated in number of wordpieces.

Table 3: The table shows, for the original dataset $\mathcal{S}$, the mean $\pm$ std of: attention $\mathbf{A}$, token attention $\mathbf{A_{ii}}$, sentence attention $\mathbf{A_{ij}}$, correlation $\rho$ and likelihood $L_s$.

| encoder | layer | $\bar{\mathbf{A}}$ | $\bar{\mathbf{A}}_{\mathbf{ii}}$ | $\bar{\mathbf{A}}_{\mathbf{ij}}$ | $\bar{L}_s$ | $\bar{\rho}$ |
|---|---|---|---|---|---|---|
| BERT | 0 | 0.099 ± 0.026 | 0.110 ± 0.028 | 0.098 ± 0.026 | 0.664 ± 0.356 | -0.146 |
| | 1 | 0.099 ± 0.026 | 0.062 ± 0.018 | 0.100 ± 0.026 | 0.664 ± 0.356 | 0.008 |
| | 2 | 0.099 ± 0.026 | 0.078 ± 0.022 | 0.099 ± 0.026 | 0.664 ± 0.356 | -0.035 |
| | 3 | 0.099 ± 0.026 | 0.047 ± 0.021 | 0.100 ± 0.026 | 0.664 ± 0.356 | -0.220 |
| | 4 | 0.099 ± 0.026 | 0.056 ± 0.023 | 0.100 ± 0.026 | 0.664 ± 0.356 | -0.236 |
| | 5 | 0.099 ± 0.026 | 0.034 ± 0.016 | 0.101 ± 0.026 | 0.664 ± 0.356 | -0.291 |
| | 6 | 0.099 ± 0.026 | 0.040 ± 0.018 | 0.100 ± 0.026 | 0.664 ± 0.356 | -0.222 |
| | 7 | 0.099 ± 0.026 | 0.038 ± 0.018 | 0.101 ± 0.026 | 0.664 ± 0.356 | -0.314 |
| | 8 | 0.099 ± 0.026 | 0.038 ± 0.017 | 0.101 ± 0.026 | 0.664 ± 0.356 | -0.150 |
| | 9 | 0.099 ± 0.026 | 0.075 ± 0.032 | 0.100 ± 0.026 | 0.664 ± 0.356 | 0.095 |
| | 10 | 0.099 ± 0.026 | 0.089 ± 0.049 | 0.100 ± 0.026 | 0.664 ± 0.356 | 0.163 |
| | 11 | 0.099 ± 0.026 | 0.070 ± 0.030 | 0.100 ± 0.026 | 0.664 ± 0.356 | 0.266 |
| RoBERTa base | 0 | 0.100 ± 0.025 | 0.158 ± 0.031 | 0.098 ± 0.025 | 0.938 ± 0.197 | 0.015 |
| | 1 | 0.100 ± 0.025 | 0.052 ± 0.029 | 0.101 ± 0.026 | 0.938 ± 0.197 | 0.027 |
| | 2 | 0.100 ± 0.025 | 0.057 ± 0.019 | 0.101 ± 0.026 | 0.938 ± 0.197 | 0.077 |
| | 3 | 0.100 ± 0.025 | 0.047 ± 0.015 | 0.101 ± 0.026 | 0.938 ± 0.197 | 0.064 |
| | 4 | 0.100 ± 0.025 | 0.052 ± 0.018 | 0.101 ± 0.025 | 0.938 ± 0.197 | -0.190 |
| | 5 | 0.100 ± 0.025 | 0.048 ± 0.017 | 0.101 ± 0.025 | 0.938 ± 0.197 | -0.014 |
| | 6 | 0.100 ± 0.025 | 0.052 ± 0.022 | 0.101 ± 0.025 | 0.938 ± 0.197 | -0.125 |
| | 7 | 0.100 ± 0.025 | 0.062 ± 0.024 | 0.100 ± 0.026 | 0.938 ± 0.197 | -0.072 |
| | 8 | 0.100 ± 0.025 | 0.045 ± 0.021 | 0.101 ± 0.026 | 0.938 ± 0.197 | -0.246 |
| | 9 | 0.100 ± 0.025 | 0.054 ± 0.035 | 0.101 ± 0.026 | 0.938 ± 0.197 | -0.275 |
| | 10 | 0.100 ± 0.025 | 0.066 ± 0.027 | 0.100 ± 0.025 | 0.938 ± 0.197 | -0.245 |
| | 11 | 0.100 ± 0.025 | 0.124 ± 0.048 | 0.099 ± 0.025 | 0.938 ± 0.197 | 0.247 |
| XLM-R | 0 | 0.089 ± 0.024 | 0.123 ± 0.039 | 0.089 ± 0.024 | 0.943 ± 0.176 | -0.088 |
| | 1 | 0.089 ± 0.024 | 0.071 ± 0.031 | 0.090 ± 0.025 | 0.943 ± 0.176 | -0.076 |
| | 2 | 0.089 ± 0.024 | 0.058 ± 0.023 | 0.090 ± 0.025 | 0.943 ± 0.176 | 0.122 |
| | 3 | 0.089 ± 0.024 | 0.075 ± 0.020 | 0.090 ± 0.025 | 0.943 ± 0.176 | 0.130 |
| | 4 | 0.089 ± 0.024 | 0.061 ± 0.018 | 0.090 ± 0.025 | 0.943 ± 0.176 | 0.075 |
| | 5 | 0.089 ± 0.024 | 0.068 ± 0.014 | 0.090 ± 0.025 | 0.943 ± 0.176 | -0.258 |
| | 6 | 0.089 ± 0.024 | 0.045 ± 0.017 | 0.091 ± 0.025 | 0.943 ± 0.176 | 0.002 |
| | 7 | 0.089 ± 0.024 | 0.054 ± 0.017 | 0.091 ± 0.025 | 0.943 ± 0.176 | -0.548 |
| | 8 | 0.089 ± 0.024 | 0.055 ± 0.019 | 0.091 ± 0.025 | 0.943 ± 0.176 | -0.407 |
| | 9 | 0.089 ± 0.024 | 0.083 ± 0.030 | 0.090 ± 0.024 | 0.943 ± 0.176 | -0.441 |
| | 10 | 0.089 ± 0.024 | 0.082 ± 0.037 | 0.090 ± 0.024 | 0.943 ± 0.176 | -0.543 |
| | 11 | 0.089 ± 0.024 | 0.132 ± 0.063 | 0.089 ± 0.024 | 0.943 ± 0.176 | 0.332 |
| RoBERTa large | 0 | 0.100 ± 0.025 | 0.199 ± 0.073 | 0.098 ± 0.024 | 0.947 ± 0.172 | -0.166 |
| | 1 | 0.100 ± 0.025 | 0.047 ± 0.030 | 0.101 ± 0.026 | 0.947 ± 0.172 | 0.180 |
| | 2 | 0.100 ± 0.025 | 0.041 ± 0.024 | 0.101 ± 0.026 | 0.947 ± 0.172 | 0.043 |
| | 3 | 0.100 ± 0.025 | 0.046 ± 0.023 | 0.101 ± 0.026 | 0.947 ± 0.172 | 0.088 |
| | 4 | 0.100 ± 0.025 | 0.053 ± 0.019 | 0.101 ± 0.025 | 0.947 ± 0.172 | -0.082 |
| | 5 | 0.100 ± 0.025 | 0.026 ± 0.015 | 0.102 ± 0.026 | 0.947 ± 0.172 | 0.008 |
| | 6 | 0.100 ± 0.025 | 0.029 ± 0.012 | 0.102 ± 0.026 | 0.947 ± 0.172 | 0.009 |
| | 7 | 0.100 ± 0.025 | 0.062 ± 0.020 | 0.101 ± 0.025 | 0.947 ± 0.172 | -0.189 |
| | 8 | 0.100 ± 0.025 | 0.079 ± 0.025 | 0.101 ± 0.025 | 0.947 ± 0.172 | -0.164 |
| | 9 | 0.100 ± 0.025 | 0.057 ± 0.018 | 0.101 ± 0.026 | 0.947 ± 0.172 | -0.121 |
| | 10 | 0.100 ± 0.025 | 0.053 ± 0.017 | 0.101 ± 0.025 | 0.947 ± 0.172 | -0.113 |
| | 11 | 0.100 ± 0.025 | 0.052 ± 0.020 | 0.101 ± 0.025 | 0.947 ± 0.172 | -0.032 |
| | 12 | 0.100 ± 0.025 | 0.052 ± 0.017 | 0.101 ± 0.025 | 0.947 ± 0.172 | -0.154 |
| | 13 | 0.100 ± 0.025 | 0.042 ± 0.021 | 0.101 ± 0.025 | 0.947 ± 0.172 | -0.048 |
| | 14 | 0.100 ± 0.025 | 0.032 ± 0.018 | 0.102 ± 0.026 | 0.947 ± 0.172 | -0.186 |
| | 15 | 0.100 ± 0.025 | 0.040 ± 0.019 | 0.101 ± 0.026 | 0.947 ± 0.172 | -0.020 |
| | 16 | 0.100 ± 0.025 | 0.042 ± 0.018 | 0.101 ± 0.026 | 0.947 ± 0.172 | -0.061 |
| | 17 | 0.100 ± 0.025 | 0.042 ± 0.019 | 0.101 ± 0.026 | 0.947 ± 0.172 | -0.013 |
| | 18 | 0.100 ± 0.025 | 0.038 ± 0.019 | 0.102 ± 0.026 | 0.947 ± 0.172 | -0.112 |
| | 19 | 0.100 ± 0.025 | 0.045 ± 0.013 | 0.101 ± 0.026 | 0.947 ± 0.172 | -0.073 |
| | 20 | 0.100 ± 0.025 | 0.055 ± 0.019 | 0.101 ± 0.026 | 0.947 ± 0.172 | -0.007 |
| | 21 | 0.100 ± 0.025 | 0.068 ± 0.034 | 0.100 ± 0.026 | 0.947 ± 0.172 | -0.195 |
| | 22 | 0.100 ± 0.025 | 0.052 ± 0.030 | 0.101 ± 0.026 | 0.947 ± 0.172 | -0.171 |
| | 23 | 0.100 ± 0.025 | 0.134 ± 0.040 | 0.099 ± 0.025 | 0.947 ± 0.172 | 0.183 |
| DistilBERT | 0 | 0.099 ± 0.026 | 0.096 ± 0.023 | 0.099 ± 0.026 | 0.785 ± 0.293 | -0.202 |
| | 1 | 0.099 ± 0.026 | 0.091 ± 0.020 | 0.099 ± 0.026 | 0.785 ± 0.293 | -0.111 |
| | 2 | 0.099 ± 0.026 | 0.045 ± 0.017 | 0.100 ± 0.026 | 0.785 ± 0.293 | -0.163 |
| | 3 | 0.099 ± 0.026 | 0.039 ± 0.018 | 0.100 ± 0.026 | 0.785 ± 0.293 | -0.185 |
| | 4 | 0.099 ± 0.026 | 0.065 ± 0.022 | 0.100 ± 0.026 | 0.785 ± 0.293 | 0.033 |
| | 5 | 0.099 ± 0.026 | 0.063 ± 0.024 | 0.100 ± 0.026 | 0.785 ± 0.293 | 0.017 |

Table 4: The table shows, for the perturbed dataset $\hat{\mathcal{S}}$, the mean $\pm$ std of: attention $\hat{\mathbf{A}}$, token attention $\hat{\mathbf{A}}_{\mathbf{ii}}$, sentence attention $\hat{\mathbf{A}}_{\mathbf{ij}}$, correlation $\hat{\rho}$ and likelihood $\hat{L}_s$.

| encoder | layer | $\bar{\mathbf{A}}$ | $\bar{\mathbf{A}}_{\mathbf{ii}}$ | $\bar{\mathbf{A}}_{\mathbf{ij}}$ | $\bar{L}_s$ | $\bar{\rho}$ |
|---|---|---|---|---|---|---|
| BERT | 0 | 0.099 ± 0.026 | 0.099 ± 0.027 | 0.099 ± 0.026 | 0.096 ± 0.198 | 0.143 |
| | 1 | 0.099 ± 0.026 | 0.050 ± 0.017 | 0.100 ± 0.026 | 0.096 ± 0.198 | 0.142 |
| | 2 | 0.099 ± 0.026 | 0.081 ± 0.023 | 0.099 ± 0.026 | 0.096 ± 0.198 | 0.447 |
| | 3 | 0.099 ± 0.026 | 0.049 ± 0.022 | 0.100 ± 0.026 | 0.096 ± 0.198 | 0.345 |
| | 4 | 0.099 ± 0.026 | 0.065 ± 0.021 | 0.100 ± 0.026 | 0.096 ± 0.198 | 0.084 |
| | 5 | 0.099 ± 0.026 | 0.051 ± 0.023 | 0.100 ± 0.026 | 0.096 ± 0.198 | 0.065 |
| | 6 | 0.099 ± 0.026 | 0.049 ± 0.023 | 0.100 ± 0.026 | 0.096 ± 0.198 | 0.353 |
| | 7 | 0.099 ± 0.026 | 0.060 ± 0.035 | 0.100 ± 0.026 | 0.096 ± 0.198 | 0.264 |
| | 8 | 0.099 ± 0.026 | 0.047 ± 0.021 | 0.100 ± 0.026 | 0.096 ± 0.198 | 0.262 |
| | 9 | 0.099 ± 0.026 | 0.063 ± 0.041 | 0.100 ± 0.026 | 0.096 ± 0.198 | 0.643 |
| | 10 | 0.099 ± 0.026 | 0.062 ± 0.052 | 0.100 ± 0.026 | 0.096 ± 0.198 | 0.588 |
| | 11 | 0.099 ± 0.026 | 0.045 ± 0.035 | 0.100 ± 0.026 | 0.096 ± 0.198 | 0.576 |
| RoBERTa base | 0 | 0.100 ± 0.025 | 0.166 ± 0.039 | 0.098 ± 0.025 | 0.072 ± 0.174 | -0.157 |
| | 1 | 0.100 ± 0.025 | 0.050 ± 0.023 | 0.101 ± 0.026 | 0.072 ± 0.174 | 0.417 |
| | 2 | 0.100 ± 0.025 | 0.050 ± 0.018 | 0.101 ± 0.026 | 0.072 ± 0.174 | 0.440 |
| | 3 | 0.100 ± 0.025 | 0.038 ± 0.018 | 0.101 ± 0.026 | 0.072 ± 0.174 | 0.488 |
| | 4 | 0.100 ± 0.025 | 0.049 ± 0.021 | 0.101 ± 0.026 | 0.072 ± 0.174 | 0.521 |
| | 5 | 0.100 ± 0.025 | 0.038 ± 0.015 | 0.101 ± 0.026 | 0.072 ± 0.174 | 0.337 |
| | 6 | 0.100 ± 0.025 | 0.064 ± 0.019 | 0.101 ± 0.025 | 0.072 ± 0.174 | -0.038 |
| | 7 | 0.100 ± 0.025 | 0.062 ± 0.028 | 0.100 ± 0.026 | 0.072 ± 0.174 | 0.544 |
| | 8 | 0.100 ± 0.025 | 0.057 ± 0.035 | 0.100 ± 0.026 | 0.072 ± 0.174 | 0.541 |
| | 9 | 0.100 ± 0.025 | 0.073 ± 0.051 | 0.100 ± 0.026 | 0.072 ± 0.174 | 0.609 |
| | 10 | 0.100 ± 0.025 | 0.071 ± 0.048 | 0.100 ± 0.026 | 0.072 ± 0.174 | 0.739 |
| | 11 | 0.100 ± 0.025 | 0.043 ± 0.033 | 0.101 ± 0.026 | 0.072 ± 0.174 | 0.802 |
| XLM-R | 0 | 0.089 ± 0.024 | 0.151 ± 0.053 | 0.088 ± 0.024 | 0.017 ± 0.071 | -0.429 |
| | 1 | 0.089 ± 0.024 | 0.116 ± 0.031 | 0.088 ± 0.024 | 0.017 ± 0.071 | 0.125 |
| | 2 | 0.089 ± 0.024 | 0.065 ± 0.022 | 0.090 ± 0.025 | 0.017 ± 0.071 | 0.283 |
| | 3 | 0.089 ± 0.024 | 0.059 ± 0.020 | 0.090 ± 0.025 | 0.017 ± 0.071 | 0.425 |
| | 4 | 0.089 ± 0.024 | 0.058 ± 0.022 | 0.090 ± 0.025 | 0.017 ± 0.071 | 0.565 |
| | 5 | 0.089 ± 0.024 | 0.058 ± 0.017 | 0.090 ± 0.025 | 0.017 ± 0.071 | 0.610 |
| | 6 | 0.089 ± 0.024 | 0.051 ± 0.021 | 0.091 ± 0.025 | 0.017 ± 0.071 | 0.516 |
| | 7 | 0.089 ± 0.024 | 0.055 ± 0.023 | 0.090 ± 0.025 | 0.017 ± 0.071 | 0.713 |
| | 8 | 0.089 ± 0.024 | 0.057 ± 0.026 | 0.090 ± 0.025 | 0.017 ± 0.071 | 0.622 |
| | 9 | 0.089 ± 0.024 | 0.095 ± 0.042 | 0.089 ± 0.025 | 0.017 ± 0.071 | 0.638 |
| | 10 | 0.089 ± 0.024 | 0.083 ± 0.050 | 0.089 ± 0.025 | 0.017 ± 0.071 | 0.709 |
| | 11 | 0.089 ± 0.024 | 0.051 ± 0.034 | 0.090 ± 0.025 | 0.017 ± 0.071 | 0.767 |
| RoBERTa large | 0 | 0.100 ± 0.025 | 0.219 ± 0.075 | 0.097 ± 0.024 | 0.055 ± 0.149 | -0.410 |
| | 1 | 0.100 ± 0.025 | 0.034 ± 0.020 | 0.101 ± 0.026 | 0.055 ± 0.149 | 0.439 |
| | 2 | 0.100 ± 0.025 | 0.036 ± 0.020 | 0.101 ± 0.026 | 0.055 ± 0.149 | 0.268 |
| | 3 | 0.100 ± 0.025 | 0.041 ± 0.018 | 0.101 ± 0.026 | 0.055 ± 0.149 | 0.441 |
| | 4 | 0.100 ± 0.025 | 0.048 ± 0.019 | 0.101 ± 0.026 | 0.055 ± 0.149 | 0.195 |
| | 5 | 0.100 ± 0.025 | 0.021 ± 0.013 | 0.102 ± 0.026 | 0.055 ± 0.149 | 0.209 |
| | 6 | 0.100 ± 0.025 | 0.031 ± 0.014 | 0.101 ± 0.026 | 0.055 ± 0.149 | 0.387 |
| | 7 | 0.100 ± 0.025 | 0.058 ± 0.022 | 0.101 ± 0.025 | 0.055 ± 0.149 | 0.214 |
| | 8 | 0.100 ± 0.025 | 0.074 ± 0.026 | 0.100 ± 0.025 | 0.055 ± 0.149 | 0.018 |
| | 9 | 0.100 ± 0.025 | 0.050 ± 0.018 | 0.101 ± 0.026 | 0.055 ± 0.149 | 0.271 |
| | 10 | 0.100 ± 0.025 | 0.048 ± 0.015 | 0.101 ± 0.026 | 0.055 ± 0.149 | 0.432 |
| | 11 | 0.100 ± 0.025 | 0.040 ± 0.017 | 0.101 ± 0.026 | 0.055 ± 0.149 | 0.156 |
| | 12 | 0.100 ± 0.025 | 0.052 ± 0.016 | 0.101 ± 0.026 | 0.055 ± 0.149 | 0.470 |
| | 13 | 0.100 ± 0.025 | 0.031 ± 0.014 | 0.102 ± 0.026 | 0.055 ± 0.149 | 0.093 |
| | 14 | 0.100 ± 0.025 | 0.030 ± 0.012 | 0.102 ± 0.026 | 0.055 ± 0.149 | 0.189 |
| | 15 | 0.100 ± 0.025 | 0.030 ± 0.012 | 0.102 ± 0.026 | 0.055 ± 0.149 | 0.407 |
| | 16 | 0.100 ± 0.025 | 0.037 ± 0.013 | 0.101 ± 0.026 | 0.055 ± 0.149 | 0.244 |
| | 17 | 0.100 ± 0.025 | 0.041 ± 0.014 | 0.101 ± 0.026 | 0.055 ± 0.149 | 0.299 |
| | 18 | 0.100 ± 0.025 | 0.042 ± 0.017 | 0.101 ± 0.026 | 0.055 ± 0.149 | 0.088 |
| | 19 | 0.100 ± 0.025 | 0.053 ± 0.020 | 0.101 ± 0.026 | 0.055 ± 0.149 | 0.534 |
| | 20 | 0.100 ± 0.025 | 0.057 ± 0.036 | 0.100 ± 0.026 | 0.055 ± 0.149 | 0.528 |
| | 21 | 0.100 ± 0.025 | 0.071 ± 0.055 | 0.100 ± 0.026 | 0.055 ± 0.149 | 0.648 |
| | 22 | 0.100 ± 0.025 | 0.056 ± 0.041 | 0.100 ± 0.026 | 0.055 ± 0.149 | 0.681 |
| | 23 | 0.100 ± 0.025 | 0.046 ± 0.035 | 0.101 ± 0.026 | 0.055 ± 0.149 | 0.823 |
| DistilBERT | 0 | 0.099 ± 0.026 | 0.086 ± 0.020 | 0.099 ± 0.026 | 0.017 ± 0.069 | -0.133 |
| | 1 | 0.099 ± 0.026 | 0.079 ± 0.019 | 0.099 ± 0.026 | 0.017 ± 0.069 | 0.489 |
| | 2 | 0.099 ± 0.026 | 0.046 ± 0.033 | 0.100 ± 0.026 | 0.017 ± 0.069 | 0.567 |
| | 3 | 0.099 ± 0.026 | 0.044 ± 0.047 | 0.100 ± 0.027 | 0.017 ± 0.069 | 0.594 |
| | 4 | 0.099 ± 0.026 | 0.045 ± 0.035 | 0.100 ± 0.026 | 0.017 ± 0.069 | 0.688 |
| | 5 | 0.099 ± 0.026 | 0.035 ± 0.033 | 0.101 ± 0.026 | 0.017 ± 0.069 | 0.558 |

