# OpenReview forum: "Attention-likelihood relationship in Transformers"
_ICLR.cc/2023/TinyPapers — Submitted to Tiny Papers @ ICLR 2023_

### Official Review · Reviewer_i5hr · 2023-03-30

**Confidence:** 4

**Summary Of Contributions:**

This paper explores the correlation between token likelihood and attention values in transformer-based language models and finds that model relies more on the context word to compute the representation of an unexpected token.

**Rating:**

Needs Clarification (NC): a submission which does not meet the reviewing criteria and needs clarification for its described problem or solution

**Strengths And Weaknesses:**

Strengths:

This paper extensively explores the correlation between token likelihood and attention values at different layers using Spearman correlation.

Weaknesses:

1. Some important details are missing:

(a) How to get “perturbed” token is unclear.

(b) The sentence attention is a vector as it is the concatenation of the other attention weights, but how can you calculate the Spearman correlation between a vector variable and a number variable?

 (c) It is unclear which type of attention is used in the calculation of the Spearman correlation between attention values and loglikelihood. (attention, token attention or sentence attention?)

2. It is unclear what the practical implications of this study are, and it appears to be a technically correct but perhaps not very useful exploration. It would be helpful if the authors could provide a more thorough explanation for the motivation and significance of their work.


**Suggested Changes:**

Typos:

1. Wrong quote in “it with a ”perturbed” token ...” in page2.

2. Tense error in appendix, such as “we retrieved all... ” -> “we retrieve all...” in page5.

---

### Official Review · Reviewer_wg13 · 2023-03-30

**Confidence:** 3

**Summary Of Contributions:**

 This paper analyzes how large language models represent a token and its attention. They find a strong correlation between token likelihood and attention values, indicating that the model relies on the token to understand its semantics.

**Rating:**

Clear, Correct, and Reproducible (CCR): a submission which meets the reviewing criteria

**Strengths And Weaknesses:**

Strength
1. The paper is well written and the problem is well defined.
2. The results are reproducible.



**Suggested Changes:**

Suggested Changes:
1. It would be beneficial if the author could provide practical examples of how their research can be applied in practical tasks.

---

### Official Review · Reviewer_1hWZ · 2023-04-01

**Confidence:** 5

**Summary Of Contributions:**

This paper empirically investigates the transformer model's attention to out-of-the-distribution words. Specifically, the authors measure transformers' attention with the original token and a perturbed token with a lower log-likelihood at position i, Lˆsi ≪ Lsi. Then, they statistically compared the attention on both the original and perturbed tokens.

**Rating:**

High Impact (HI): a submission which meets the reviewing criteria and is predicted to make an impact on the field

**Strengths And Weaknesses:**

Strength:
Systematic investigation of the transformer's attention for the out-of-distribution words.

Weakness
Although the perturbed attention showed a stong correlation ρ, the correlation was not statistically significant as demonstrated by the p-value

**Suggested Changes:**

I enjoyed reading the paper.

The authors could investigate a follow-up study along this line: how does the model's size impact the out-of-distribution tokens' attention?
Do LLMs with large parameter sizes handle the out-of-distribution tokens well than the smaller ones?
From the table, the RoBERTa large has the greatest correlation coefficient.

---

### Meta-Review · Area_Chair_2qGW · 2023-04-06

**Recommendation:** Invite to present
**Confidence:** 4

**Metareview:**

This paper explores the correlation between token likelihood and attention values at different layers using Spearman correlation.
The paper is well-written, the experiments support the hypothesis and the authors followed the submission guidelines.
There are some clarifications needed for instance the authors mention that the sentence attention is a vector as it is the concatenation of the other attention weights, but it's not clear how they calculate the Spearman correlation between a vector variable and a number variable.
Overall, this is a nice submission with valued findings and high impact to the field.

**Summary:**

Investigates token likelihood correlation with attention in transformers. Shows strong correlation, even though the explanation of the experimental settings is not entirely clear.

**Comments And Feedback To The Authors:**

This paper explores the correlation between token likelihood and attention values at different layers using Spearman correlation.
The paper is well-written, the experiments support the hypothesis and the authors followed the submission guidelines.
There are some clarifications needed for instance the authors mention that the sentence attention is a vector as it is the concatenation of the other attention weights, but it's not clear how they calculate the Spearman correlation between a vector variable and a number variable.
Overall, this is a nice submission with valued findings and high impact to the field.

**Reason For Not Giving A Higher Recommendation:**

The reviewers correctly asked for some clarification in the experimental settings.

**Reason For Not Giving A Lower Recommendation:**

While clarification is important to understand the message of the paper, we hope in the revised version it is addressed since the idea is novel and valuable and the paper is otherwise well-presented.

---

### Decision · Program_Chairs · 2023-04-07

Invite to present